# An Exploratory Study of Pandemic-Restricted Travel—A New Form of Travel Pattern on the during- and Post-COVID-19 Era

**DOI:** 10.3390/ijerph19074149

**Published:** 2022-03-31

**Authors:** Shan Wang, Ivan Ka-Wai Lai, Jose Weng-Chou Wong

**Affiliations:** 1Faculty of Hospitality and Tourism Management, Macau University of Science and Technology, Macau 999078, China; wangsh89@outlook.com; 2Faculty of International Tourism and Management, City University of Macau, Macau 999078, China; ivanlai@cityu.mo

**Keywords:** pandemic-restricted travel, motivation to travel, constraint to normal travel, satisfaction, intention of continuous travel intention, prospect theory, PLS-SEM

## Abstract

This study introduces a new travel pattern “pandemic-restricted travel” that exists from COVID-19 based on prospect theory. The purpose of this study is to incorporate the motivation to travel and constraint to normal travel to predict tourists’ intention to continue visiting other alternative destinations due to COVID-19 restrictions. This study first generated the items of motivation to travel and constraints to normal travel from a focus group interview with 15 travel industry professionals in December 2020 in Zhuhai. Then, an online survey collected data from 416 respondents in the Greater Bay Area of China from January to February 2021. The results of exploratory factor analysis using SPSS identified two factors of motivation to travel (leisure and exploration) and two factors of favourable constraints to normal travel (policy restriction and perceived risk). The results of partial least squares–structural equation modelling (PLS-SEM) indicated that these four factors positively influence satisfaction but only leisure and exploration factors positively influence the intention of continuous pandemic-restricted travel. Among the four factors, leisure has the strongest impact on both satisfaction and intention of continue travelling. The results also revealed that satisfaction fully mediates the effects of two constraint factors and partially mediates the effects of two motivation factors on the intention of continuous pandemic-restricted travel. Implications for researchers and governments for pandemic-restricted travel during and in the post-COVID-19 era are then discussed.

## 1. Introduction

COVID-19 has spread rapidly and severely impacted the tourism industry [1]. Up to 2021, most countries continuously implemented quarantine policies to prevent the epidemic situation [2]. While the demand for tourists still exists, the form of travel will be changed and travelling under the effects of COVID-19 will become a new normal [3]. Thus, it is necessary to extend our research focus from crisis management and recovery during COVID-19 to the new travel format during and after the COVID-19. 

Although COVID-19 has not been fully controlled, many destinations are attempting to recover tourism to respond to the tourists’ needs as well as to restore their tourism economy [4]. For instance, recently, some countries have started establishing the travel bubble strategy that allows citizens to travel freely between specified nations [5]. However, with the different policies among countries and tourists’ perceived risks on COVID-19, the implementation of the travel bubble faces many issues [6]. On the other hand, Wen et al. [7] made a summary that independent tourism and nature tourism may lead future travel patterns since people have more preference to stay in a natural environment and with fewer group people. Bae and Chang [8] emphasized the importance of “untact tourism”, which mainly focuses on physical interaction and social distancing to decrease people’s risk perception towards COVID-19. Current studies provided different concepts of future travel patterns but have not studied the travellers’ attitudes and behavioural intention towards these travel patterns. According to Kahneman and Tversky’s prospect theory [9], individuals’ travel preferences and behaviour under risky conditions depend on their evaluation of loss and gains of alternative choices [10]. Therefore, this study introduces an alternative choice, the concept of “pandemic-restricted travel”, that fulfils both tourists’ demand and reduce risk perception. By substituting the normal travel, the pandemic-restricted travel, especially the tourist’s perceptions of this travel pattern, is worthy of being studied under COVID-19.

Previous studies have identified different motivations (e.g., enjoyment) and constraints (e.g., perceived risk) for normal travel [11]. However, pandemic-restricted travel as a way of inverting-selection tourism, its motivations and constraints should be different from normal travel. In previous studies, researchers consider a constraint as a negative motivation to go to a certain place. However, due to several restrictions and risks for normal travel, a constraint to normal travel becomes a favourable constraint, a passive motivation to take pandemic-restricted travel. Therefore, to understand pandemic-restricted travel, it is necessary to examine the motivations for new travel and constraints to normal travel in the era of COVID-19.

The aim of the study is to investigate the effect of motivations to travel and constraints to normal travel on satisfaction with and the intention to continue pandemic-restricted travel. To measure the constraints to normal travel under COVID-19, new measurable items should be tailor developed. Therefore, this research contributes to developing the measurement scales of motivations to travel and constraints to normal travel under COVID-19. The results of the study enrich the literature by providing an understanding of how these two factors influence people’s intention to continue pandemic-restricted travel. This study is also the first to explore the (favourable) constraint as passive motivation in the era of COVID-19. Practically, this research provides several implications for local governments and Destination Management Organizations (DMOs) to formulate marketing strategies and enhance the cooperation of various stakeholders in cross-region under COVID-19.

## 2. Theoretical Background

### 2.1. The Concept of Pandemic-Restricted Travel

A variety of studies have explored the impact of COVID-19 on tourism [1]. For instance, Shamshiripour et al. [12] discussed the change in people’s travel behaviour in COVID-19 and revealed that people’s mobility styles and habitual travel behaviour have significantly changed. To explore changes in travel behaviour during COVID-19, researchers have tested the effect of risk perception and social influence on going-out-restriction and found that it is plausible to find an alternative way for shopping and travel, instead of reducing frequencies [13]. From these previous findings, it can be acknowledged that COVID-19 had changed the number and types of outdoor activities. However, the future travel pattern is still waiting to explore.

Since COVID-19 has dramatically changed tourists’ psychological attitudes [14], the travel pattern becomes selective, and some projected tourism trends may drive future tourism, such as independent travel, luxury trips and health and wellness tourism [7]. Although researchers have suggested some possible tourism patterns [15], they have not considered the travel restrictions that influence tourists’ motivation and perception to travel [16]. Although COVID-19 vaccines are probably one of the best ways to help the tourism economy recover, it is not a “free pass”. Vaccination for travel depends on many factors [17]. Vaccinated people may spread COVID-19 [18]; therefore, people are not free to travel shortly. 

Therefore, the new form of tourism with a regional scope completely fulfils tourists’ demand and reduces more risky conditions. This new travel pattern may lead to the future tourism landscape, which is named “pandemic-restricted travel”. As no existing literature has mentioned this type of travel pattern, inspired by Pizam et al.’s [19] description of travel, the working definition of pandemic-restricted travel in this study is defined as people’s participation in touristic activities under the restricted conditions of the COVID-19 pandemic.

### 2.2. Motivation of Pandemic-Restricted Travel and Constraint to Normal Travel

Motivation is a psychological need to integrate a person’s behaviour [20]. In tourism research, travel motivation influences tourists’ travel behaviour and travel choice [21]. There are different types of motivations, such as different cultures, intellectual improvement, new places, relaxation, pleasure, and entertainment [22].

On the other hand, many tourism studies of motivation include the constraint as well [11]. Travel constraint refers to multiple factors of inability to travel and preventing people from continuing travel [20]. For example, Chen et al. [23] examined unfamiliar culture as a constraint in visiting a destination, and Huang and Hsu [24] revealed disinterest as a constraint in travelling to Hong Kong. Indeed, these studies identified factors of constraint as a negative stimulus and suggested several strategies to minimize travel constraints.

The prospect theory was originally developed by Kahneman and Tversky [9], combining economics and psychological states to explain people’s decision making under uncertain or risky events. The prospect theory can be applied to explain why an individual makes the choices when facing risky conditions [25]. Recent tourism studies have applied the prospect theory to explore tourists’ perception of revisiting a destination that has certain risks [26]. This theory implies that when people make the decision to travel in a risky condition, they will evaluate the perceived loss and gain from the travel. If the level of gain from travel is the same, they will prefer a form of travel in which the level of perceived loss is lower. In the era of COVID-19, since many regions have precaution policies that have restrictions to travel, people’s normal travel preference is limited [2]. Furthermore, risk perception significantly influences tourists’ decision making in tourism choice [27]. Thus, the prospect theory is appropriate with the condition of COVID-19, in which extreme uncertainty generates fear [28]. After considering the policy restriction and perceived risk for normal travel, people’s constraint to normal travel becomes a passive motivation for them taking pandemic-restricted travel. However, research on passive travel motivation is still limited and the understanding of favourable constraint as a positive stimulus should be explored. Consequently, this study combines active motivation and passive motivation in the COVID-19 setting.

### 2.3. Satisfaction

Satisfaction is defined as “the interaction between a tourism experience and expectation about the destination” [29] (p. 315). A close relationship between motivation and satisfaction has been empirically investigated in previous studies. For instance, Agyeiwaah et al. [30] showed that culinary tourists’ motivation positively influence satisfaction based on an analysis of 300 international tourists at cooking schools in Thailand. In the era of COVID-19, satisfaction is highly important to reflect tourism experience and it is a key strategy to recover tourism successfully [31]. Previous studies revealed that motivation can generate tourists’ expectations, which, in turn, satisfy people’s tourism experience. In addition, constraint to normal travel, as what has been defined in this study, refers to the way of passive motivation that stimulates people with physiological feelings and positive attitudes towards pandemic-restricted travel. Therefore, the constraint to normal travel as passive motivation may produce a positive outcome of pandemic-restricted travel. Therefore, this study presents the following hypotheses:

**Hypotheses** **1** **(H1).**
*Motivation positively influences the satisfaction with pandemic-restricted travel.*


**Hypotheses** **2** **(H2).**
*Constraints to normal travel positively influence the satisfaction with pandemic-restricted travel.*


### 2.4. The Intention of Continuous Visiting of Other Alternative Destinations Due to COVID-19 Restrictions

Revisit intention was described as “revisit intention is an extension and antecedent of tourist overall satisfaction” [32] (p. 1142). In the existing literature on the satisfaction effect on repeating visitation, many other studies have found a causal relationship between them [33]. Under the background of COVID-19, Kim et al. [34] indicated that satisfaction significantly influences revisit intention towards South Korean domestic tourism. 

However, regarding COVID-19, the concept of revisit intention should be extended in this study, which should target multiple nearby destinations on helping tourism recovery. Wong et al. [35] in the ethnic minority tourism context examined tourists’ behavioural intention to other ethnic destinations. It is an intention to continuous ethnic tourism. In this study, the target of behavioural intention is continuous pandemic-restricted travel, which corresponds with that of revisit intention. The intention of continuous visiting of other destinations in this study refers to visiting tourist destinations with a close distance, which is a more suitable topic with the pandemic-restricted travel context. Therefore, if a tourist is satisfied with pandemic-restricted travel, he/she may like to take pandemic-restricted travel to other closed destinations in the future. Hence, this study presents the following hypothesis: 

**Hypotheses** **3** **(H3).**
*Satisfaction with pandemic-restricted travel positively influences the intention of continuous pandemic-restricted travel.*


As Lee et al. [36] describe, ecotourists’ intention to revisit restored ecological parks is influenced by motivation factors. Similarly, as discussed in the above literature, although Huang and Hsu [24] explored the negative effect of constraint on revisit intention, it is argued that the function of favourable constraint is different from a normal constraint, which generates a positive effect on tourist behaviour. Replacing revisit intention with the intention of continuous pandemic-restricted travel, the motivation to travel and constraint to normal travel may be predictors that influence visiting other pandemic-restricted travel destinations in the era of COVID-19. Therefore, the following hypotheses are presented.

**Hypotheses** **4** **(H4).**
*Motivation positively influences the intention of continuous pandemic-restricted travel.*


**Hypotheses** **5** **(H5).**
*Constraints to normal travel positively influence the intention of continuous pandemic-restricted travel.*


## 3. Research Method

### 3.1. Research Framework

According to the discussion above, the motivation to take pandemic-restricted travel involves two categories (motivation and constraint to normal travel) under the background of pandemic-restricted travel. Previous studies have constructed a “motivation-perception-behavioural intention” model to examine the relationships between motivation, satisfaction, and behavioural intention [37,38]. For example, He and Luo [38] explored the effect of ski tourism motivation on visitors’ satisfaction and revisit intentions. Therefore, this study constructs a conceptual model as shown in Figure 1 that examine the effect of motivation to take pandemic-restricted travel and constraint to normal travel on satisfaction with pandemic-restricted travel and the intention of continuous pandemic-restricted travel.

### 3.2. Instrument Development

Since there are no existing scales that can be directly applied in the pandemic-restricted travel under COVID-19, this study adopted a qualitative approach to derive items of motivation to travel and constraint to normal travel in the context of pandemic-restricted travel. A focus group interview was performed with 15 travel industry professionals in the Guangdong Province of China in December 2020. To facilitate the discussion in the interviews, the measurement scales of motivation from Khan et al. [22] and Suhartanto et al. [39] and the items of constraint from Lai et al. [40] and Huang and Hsu [24] are used as references. Based on their measurement items, interviewees were asked to select or create the key motives for pandemic-restricted travel within the province and the key factors that restrict tourists’ willingness to normal travel (include other provinces and other countries). The results from interviews formulated 10 items of constraint and 11 items of motivation. In total, two items of motivation and five items of constraint were newly added. The suggested items of motivation to travel are roughly similar to the previous research; for example, experience different ways of life, accompany families and friends, and increase knowledge. Most recommended items of constraint to normal travel are newly built based on the status quo, such as restrictions by the workplace and government policy. Additionally, some items of constraint from previous studies were not included because these items are not appropriate in the situations of pandemic-restricted travel, like the language barrier, do not have enough holidays, and few travel agencies.

### 3.3. Measurements

Besides the measurements of motivation and constraint, the continuous visiting of other pandemic-restricted travel destinations is measured by revising a three-item scale from Wong et al. [35]. For example, “I would visit other pandemic-restricted travel destinations in the future.” Four items of satisfaction with pandemic-restricted travel are measured by using Lai and Hitchcock [41]. For example, “Overall, I am fully satisfied with tourism experiences on this trip”. After the item generation, four university professors in the field of hospitality and tourism were invited to confirm the accuracy of these measurement items and no revision had been made. 

The preliminary questionnaire was designed in English and then translated into Chinese by a professional translator. Another professional translator was invited to make a back-translation. After confirming no translation errors, the final version of the Chinese questionnaire was ready for the pilot test. 

### 3.4. Data Collection

In this study, the questionnaire consists of four sections. The first section includes the screening question “Have you ever travelled in your province in the past three months?”. Respondents who did not have travel experience in the last three months in the same province were not invited to fill out the survey. The second section includes measurement scales of motivation and constraint. The third section includes socio-demographic information, while the last section involves measurement items of satisfaction and the intention of continuous pandemic-restricted travel. A pilot study was conducted with 50 residents in Zhuhai, Guangdong, China. The participants did not indicate having any problems understanding the questionnaire. Following government guidance on social distancing to protect research students, it is difficult to conduct a face-to-face survey. Therefore, an online survey was the most effective way to prevent social distancing and recruit participation [42]. The data were collected by convenience sampling method. The online survey was performed by using Sojump [43], one of the most popular and widely used online survey tools in China [44,45]. Considering that the content of the present study refers to travel in the same region or province, respondents of pandemic-restricted travel are geographically limited to the Greater Bay Area (GBA). To ensure the diversity of the sample, all questionnaires were set to the IP address before distributing in the GBA, and the online questionnaires with the web link were formally sent to participants in the GBA. The GBA has a total of 69 million residents with a US $1.5 trillion GDP [46]. Because each city of the GBA has its unique tourism resources, the GBA is a novel travel experience for tourists [31]. No incentives were provided. Finally, a total of 475 participants finished the survey from January to February 2021. Of the 475 responses, 59 responses containing the most similar rates were extracted from the survey, and finally, 416 responses were successfully applied for data analysis. The sample characteristics are shown in Table 1.

## 4. Results

### 4.1. Descriptive Statistics of Measurement Scales

Table 2 shows the descriptive statistics of the measurement scales, including mean, standard deviation, excess kurtosis, and skewness. The values of excess kurtosis and skewness are greater than −3.0 and less than 3.0, so the data are relative to the normal distribution.

### 4.2. Exploratory Factor Analysis (EFA)

Following Lai and Hitchcock [47], exploratory factor analysis (EFA) with a principal component analysis and varimax rotation method was conducted to extract dimensions. Data were analysed by SPSS software. Firstly, 11 items of motivation to travel were tested. The initial Kaiser-Meyer-Olkin (KMO) is 0.856. After one cycle reduction, two items (accompany families and friends, destination promotion) were deleted, and therefore, nine items for two dimensions of motivation were categorized as exploration (four items) and leisure (five items). The final KMO is 0.858. All nine items show acceptable loadings while the lowest value is 0.558 (>0.50) [48]. They occupy 58.66% of the total variance. Similarly, an EFA was conducted on the 10 items of constraint. The initial KMO value is 0.873. After one cycle reduction, two items (high risk of long-distance trips and unknown policies of other provinces) were removed. The two dimensions are extracted and renamed as policy restriction (three items) and perceived risk for normal travel (five items). The final KMO is 0.848. The lowest factor loading is 0.640. They occupy 57.127% of the total variance. As such, Table 3 shows the results of EFA.

### 4.3. Measurement Model Evaluation

In the present study, partial least squares structural equation modelling (PLS-SEM) was used to evaluate the measurement model and structural model. Following Hair et al. [49], PLS-SEM has an advantage on relatively small or medium size sampling, and it focuses on maximizing the variance of the dependent variables that are explained by the independent variables. Different from covariance-based SEM (CB-SEM), it mainly focuses on theory tests [49]. This study focuses on the prediction of the dependent variable; thus, it is considered PLS-SEM is more appropriate.

Confirmatory factor analysis (CFA) was conducted sequentially by SmartPLS v. 3.3.3 [49] to examine factor structure. According to Hair et al. [48], the factor loading higher than 0.7 is regarded as the recommended level. However, one factor of motivation (close to my home) and one factor of constraint (decrease of disposable income) are with an unacceptable factor loading (0.639 and 0.514, respectively) in the first round of CFA. Therefore, these two items were removed. Then, all factor loadings are higher than 0.7 as shown in Figure 1. As shown in Table 4, the values of Cronbach’s alpha and construct reliability (CR) are all higher than the accepted level of 0.7, and the values of the average variance extracted (AVE) are all greater than 0.5 as the recommended level. Therefore, there are no issues about convergent validity and internal consistency of the measurement variables. The discriminate validity was also confirmed since the square root of the AVE of each factor exceeds the correlations between potential variables [50] and all values of the Heterotrait-Monotrait ratio (HTMT) were lower than 0.85 [49], indicating the acceptable level of discriminate validity. Therefore, the measurement scales are valid and reliable for structural model evaluation.

### 4.4. Structural Model Evaluation

To evaluate the proposed model, this study used structural equation analysis with 416 valid samples on bootstrapping with 5000 subsamples. As Hair et al. [49] recommend, the analysis results of several indicators (path coefficient, *p*-value, and VIF) are measured in Table 5. As the EFA results, hypotheses for motivation and constraint were divided into two as (a) and (b) to represent the two dimensions. The two dimensions of motivation all significantly related to satisfaction with pandemic-restricted travel (H1a: ß = 0.167, *p* < 0.01; H1b: ß = 0.410, *p* < 0.001) and the intention of continuous travel (H4a: ß = 0.109, *p* < 0.05; H4b: ß = 0.206, *p* < 0.001), while the two dimensions of constraint also significantly explained satisfaction with pandemic-restricted travel (H2a: ß = 0.146, *p* < 0.01; H2b: ß = 0.138, *p* < 0.05) but not significantly explained the intention of continuous travel (H5a: ß = 0.062, *p* > 0.05; H5b: ß = 0.063, *p* > 0.05). The last hypothesis was supported as there was a significant relationship between satisfaction and the intention of continuous travel (H3: ß = 0.310, *p* < 0.001). The results suggest that all hypotheses were supported except H5a and H5b.

As the most important criterion, R-square predicts endogenous variables and its value greater than 0.25 indicates the strong power [48]. Accordingly, the R-square of satisfaction was 0.446 and that of the intention of continuous visiting was 0.355, which demonstrated the sufficient power of the proposed model (see Figure 2).

### 4.5. Mediating Effects Test

To deeply understand the indirect and total effects of motivations to travel and constraints to normal travel on the intention of continuous pandemic-restricted travel, this study examined specific indirect effects analysis with 5000 subsamples and a 95% confidence interval. The results as shown in Table 6 indicated that satisfaction mediates the relationship between two factors of motivation (exploration and leisure) and the intention of continuous pandemic-restricted travel (Exploration: indirect effect = 0.052, *p* = 0.017 < 0.05, 2.5% interval = 0.016 > 0; Leisure: indirect effect = 0.127, *p* = 0.000 < 0.001, 2.5% interval = 0.073 > 0). The total effects of leisure and exploration on the intention of continuous pandemic-restricted travel are with a *p*-value of 0.000 (*p* < 0.001) and 0.002 (*p* < 0.01). The variance accounted for (VAF) values of leisure and exploration are 0.381 and 0.323, respectively. It is regarded as partial mediation, since VAF values are between 20% and 80% [48]. In addition, satisfaction mediates the relationship between two factors of constraint (policy restriction and perceived risk) and the intention of continuous pandemic-restricted travel (Policy restriction: indirect effect = 0.045, *p* = 0.015 < 0.05, 2.5% interval = 0.013 > 0; Perceived risk: indirect effect = 0.043, *p* = 0.029 <0.05, 2.5% interval = 0.008 > 0). However, the direct effects of policy restriction and perceived risk on the intention of continuous pandemic-restricted travel are not significant (*p* > 0.05). Therefore, satisfaction fully mediates the effects of two constraint factors (policy restriction and perceived risk) and partially mediates the effects of two motivation factors (leisure and exploration) on the intention of continuous pandemic-restricted travel.

## 5. Discussion

The results show that two motivation factors positively influence satisfaction and intention of continuous pandemic-restricted travel. Between these two factors, leisure is the major dominator to influence satisfaction and intention to take continuous pandemic-restricted travel. These results are in line with previous studies in traditional tourism that travel motivation is positively related to travel satisfaction [51] and the intention to visit [52]. In traditional tourism, motivation could include many factors such as novelty, health, relaxation, socialization, self-actualization, and nostalgia [51,52]. However, motivation to pandemic-restricted travel mainly includes leisure and exploration. On the other hand, two factors of constraint to normal travel (policy restriction and perceived risk) positively influence tourists’ satisfaction with pandemic-restricted travel. This is contrary to Pan et al. [51], which state that constraint factors such as external resources, time, lack of companionship do not significantly affect travel satisfaction. It implies that the effects of travel constraint factors of traditional tourism are less important compared with motivation factors. However, during the COVID-19 pandemic, according to the prospect theory, the constraint in the present study is regarded as the invert-selection motivation, so the effect of which is significant. Both of the constraints (perceived risk and policy restriction) do not directly influence the intention of continuous pandemic-restricted travel. These results are similar to the study of Lee et al. [53], which argued that three travel constraints, including intrinsic, interactional, and environmental constraints, do not directly influence intention to travel, but influence it indirectly through helplessness in disabled tourism. It may imply that the invert-selection motivation may not work as an active motivation direct drive to travel.

The results also show that there is a positive relationship between satisfaction and the intention of continuous pandemic-restricted travel. Although many previous studies have found a positive relationship between travel satisfaction and intention of revisit [54], studying the effect of satisfaction on the visit intention to similar destinations was rare. Furthermore, satisfaction partially mediates the relationship between two factors (exploration and leisure) of motivation to travel and the intention of continuous pandemic-restricted travel and fully mediates the relationship between two factors of constraint to normal travel (policy restriction and perceived risk) and the intention of continuous pandemic-restricted travel. This is different from prior research, which showed only the (partial [37] and full [30]) mediating role of satisfaction on the relationship between motivation (as a single factor) and behavioural intention but not in detail. This study shows that leisure has the greatest total effect on the intention of continuous pandemic-restricted travel (total effect = 0.334), followed by exploration (total effect = 0.166). People continue taking pandemic-restricted travel because they are looking for relaxation and shopping as well as for knowing different cultures and places. Policy restriction is the major factor of the constraint to normal tourism affecting the intention of continuous pandemic-restricted travel (total effect = 0.118), but not the perceived risk of normal tourism (*p*-value > 0.05). People are concerned about policy restrictions such as quarantine but do not worry about crowded places and long-distance trips when considering the continuous pandemic-restricted travel. 

### 5.1. Theoretical Implications

Firstly, different researchers have different focuses and argued different travel patterns (e.g., [7,8]). For overcoming the limitation in generalizing the results in a particular type of during- or post-COVID travel, this study introduces the concept of pandemic-restricted travel that provides researchers with a holistic view of further travel patterns. Then, based on the prospect theory, this study develops the research framework of pandemic-restricted travel. Therefore, this study contributes to the during- and post-COVID-19 travel research that unifies existing during- or post-COVID travel patterns. Because of what these travel patterns have in common, they have a common influence on tourists. Therefore, this study helps researchers to examine tourists’ attributes and behaviours toward this new normal travel.

Secondly, this study contributes to travel motivation research by classifying the travel constraint as an invert-selection motivation according to the prospect theory. People will choose an option with smaller losses under a risky condition. This concept is also supported by the game theory that people usually choose the more favourable one when they are facing to choose between two options [55]. Previous studies in travel motivation seldom considered the existence of an alternative option. It implies that in order to study certain types of travel, researchers should consider invert-selection motivation in their future research. On the other hand, although the travel motivation factors explored in this study are tailored for pandemic-restricted travel, they may also be applied for a certain type of travel. Furthermore, researchers can compare the effects of motivation factors (as active motivation factors) with favourable constraint factors (as passive motivation factors) on taking travel.

Thirdly, this study contributes to during- and post-COVID-19 tourism research by developing measurement scales for motivation to travel and constraint to normal travel by a focus group interview. The results of exploratory factor analysis classified two factors of motivation and two factors of constraint. The motivation to travel during the pandemic is mainly for leisure and exploration. The constraint to normal travel consists of internal and external constraints. Policy restriction is an external constraint that is controlled by governments. During the pandemic, the governments enacted a series of restrictions that are unavoidable for ordinary tourists. Perceived risk is an internal constraint that affects tourists’ psychological fear of being infected. This psychological barrier is not limited to normal travel. Even for pandemic-restricted travel, where tourists travel within the province, they still should have concerns of prevention practices, such as wearing masks, hand washing, taking public transportation avoiding crowded places [56]. Therefore, the developed measurement scale of constraint to normal travel can be applied for studying any type of pandemic-restricted travel. This study enriches our understanding of motivation and the favourable constraint during COVID-19. These measurement scales can provide references for researchers to conduct their COVID-19 tourism research.

Fourthly, since the invert-selection motivation is not generated from tourists’ psychological needs, it is counted as a passive motivation. This study found that satisfaction serves as a partial mediator on active motivation (generated from tourists’ psychological needs) and a full mediator on passive motivation. It implies that cognitive attitudes actively generated affect affective attitudes and behaviours, while cognitive attitudes passively generated only affect affective attitudes and do not directly affect behaviours. This inference can explain why satisfaction in some studies plays a partial mediating role, while in some acts as full mediators. For example, Bayih and Singh [37] examined the partial mediating role of satisfaction between motivation and revisit intention in domestic tourism, and satisfaction plays the fully mediating role between motivation and loyalty in culinary tourism [30]. Undoubtfully, this inference needs more research to confirm in the future. This study preliminarily suggests that the partial or full mediating effect of satisfaction depends on the nature of its antecedents. Therefore, researchers should consider this when designing their research model for studying tourist satisfaction.

Lastly, this study also highlights the mechanism that influences tourists to consider further pandemic-restricted travel in other destinations. Many studies have emphasized the importance of repeat behaviours and have taken revisit intention as a major outcome [52]. However, people seldom go to the same tourist destination after a short period, so researchers recently proposed testing the visit to similar destinations [35]. This study shows that (active and passive) motivations influence tourist satisfaction with a form of alternative travel (pandemic-restricted travel) that leads to continuing that travelling pattern in the future in multiple destinations. It supports researchers to examine the intention of continuous travel patterns instead of only revisiting the same tourist destination.

### 5.2. Practical Implications

Based on the findings, there are some practical recommendations for regional governments and tourism businesses during and post-COVID-19. In pandemic-restricted travel, what travellers demand are leisure and exploration, so destinations should focus on building leisure and novel facilities. For example, tourism enterprises should analyse tourists’ preference for travelling to a destination and develop a variety of tourism activities, which enable tourists to experience different ways of life and fresh activities. In addition, the new attractions could be developed, so that people have more choice to explore new places and increase their knowledge. In addition, to promote pandemic-restricted travel, the regional governments could waive the entry fee for the attraction points for local people. Furthermore, the local governments could cooperate with tourism enterprises such as hotels to issue promotion programs such as buy one get one free to stimulate the travel demand. At the same time, the local governments should also pay attention to risk management and pandemic prevention.

In the past, large tourism enterprises have the resources to make large investments in infrastructure such as large shopping malls to attract foreign tourists. Considering the constraint to normal travel, pandemic-restricted travel is an opportunity for small- and medium-sized companies because travellers are locals and nearby people. They avoid being crowded with a large group of people and are more willing to explore local boutique stores and special places. Therefore, small- and medium-sized enterprises could implement diversification strategies to attract nearby tourists and advertise their products on social media platforms, such as a WeChat Public Account, hashtag hot topics on Instagram and TikTok.

### 5.3. Limitations and Future Studies

This study contains several limitations which should be addressed in future research. Firstly, the major limitation is samples of this study came from the GBA, including nine cities of Guangdong province. As pandemic-restricted travel is considered a growing opportunity in the era of COVID-19, findings and conclusions may not have a generalization, and future research could target other countries and regions. A comparison could also be considered between China and overseas. Secondly, this study examined people’s future intention whether they want continuous visit other pandemic-restricted travel destinations. Future studies can consider integrating actual behaviour in the model. Thirdly, the scales in this study were applied for pandemic-restricted travel. Because of the long-term effect and unknown period of COVID-19, scales of motivation to travel and constraint to normal travel may change. Future studies should attempt to verify and update the scales until the travel is completely recovered. Finally, in this study, there is an assumption in the temporal order of the *motivation* and *constraint* to *satisfaction* to the *continuous pandemic-restricted travel*. There is a limitation in the causal influence of the *satisfaction* to the *continuous pandemic-restricted travel* because some unmeasured confounding variables may cause this causal relationship. Further studies are recommended to explore any confounding variables.

## 6. Conclusions

Under the great impacts of COVID-19 on the tourism industry, the new travel pattern “pandemic-restricted travel” is proposed, and the concept of (favourable) constraint to normal travel as passive motivation is introduced. This study aims to investigate the relationships among motivation to travel (leisure and exploration), favourable constraint to normal travel (policy restriction and perceived risk), satisfaction with the pandemic-restricted travel, and the intention of continuous pandemic-restricted travel. The Greater Bay Area of China was chosen as the research site to perform the focus group interview and survey. The results indicate that all two factors of motivation and two factors of favourable constraint affect satisfaction with pandemic-restricted travel, but only two factors of motivation can directly affect the intention of continuous pandemic-restricted travel. Leisure is the key among the four factors affecting all outcome variables, and satisfaction takes a mediating role in these relationships. Overall, this research provides a research framework of the new travel pattern during and in the post-COVID-19 era.

## Figures and Tables

**Figure 1 ijerph-19-04149-f001:**
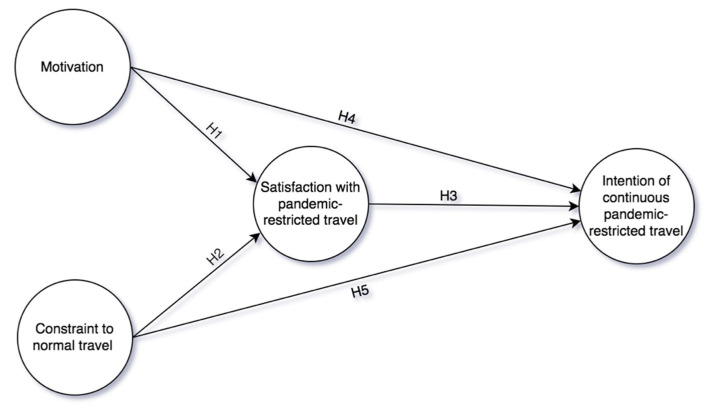
Proposed model.

**Figure 2 ijerph-19-04149-f002:**
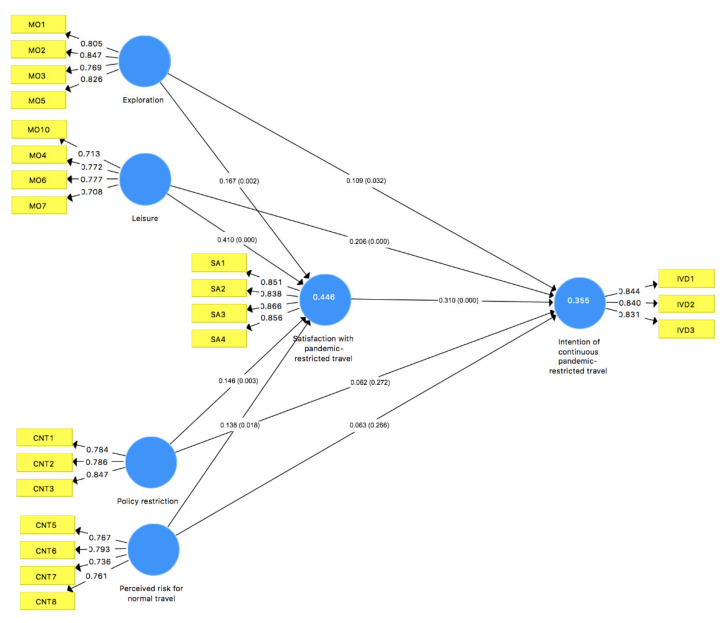
Results of PLS-SEM analysis.

**Table 1 ijerph-19-04149-t001:** Demographic profile (*n* = 416).

Variable	*n*	%		*n*	%
Gender			Education level		
Female	217	52.2	Middle school	4	1.0
Male	199	47.8	High school	15	3.6
Age			College	37	8.9
18–20	16	3.8	Undergraduate	278	66.8
21–25	149	35.8	Postgraduate and above	82	19.7
26–30	127	30.5	Monthly Income (USD)		
31–35	62	14.9	430 or below	102	24.5
36–40	29	7.0	431–720	54	13.0
41–45	14	3.4	721–1145	113	27.2
46–50	10	2.4	1146–1430	62	14.9
Above 51	9	2.2	1431–2145	57	13.7
Marital status			Over 2145	28	6.7
Single	255	61.3			
Married	161	38.7			

**Table 2 ijerph-19-04149-t002:** Descriptive statistics of measurement scales.

		Mean	Standard Deviation	Excess Kurtosis	Skewness
Motivation to travel					
MO1	To know different cultures/ways of life.	5.099	1.167	0.344	−0.384
MO2	To travel for intellectual improvement.	5.264	1.216	0.020	−0.421
MO3	To travel to know new, different places.	5.387	1.161	−0.206	−0.395
MO4	To travel to rest and relaxation purposes.	5.825	1.079	1.034	−0.845
MO5	To travel to seek adventure and pleasure.	5.243	1.193	0.390	−0.512
MO6 ^%^	To travel to seek gastronomy.	5.548	1.163	0.546	−0.622
MO7	To travel to go shopping.	4.942	1.309	−0.399	−0.209
MO8 *	To travel to accompany friends/families.	4.993	1.542	−0.001	−0.734
MO9 ^#%^	It is close to my place of residence.	5.200	1.301	0.207	−0.611
MO10	Because it is a tourist destination that suits my budget.	5.421	1.192	1.159	−0.806
MO11 *	To travel attracted by promotion.	3.945	1.527	−0.361	0.106
Constraint to normal travel					
CNT1	Getting travel documents to other provinces during COVID-19 is not easy.	5.490	1.353	1.034	−0.975
CNT2 ^%^	Work unit/school has policy restrictions (stay in the province).	5.344	1.359	0.506	−0.767
CNT3 ^%^	The provincial government has policy restrictions from other provinces, such as quarantine.	5.675	1.276	1.133	-1.019
CNT4 *	The risk of a long-distance trip is high.	5.743	1.166	0.923	−0.922
CNT5 ^#^	The risk of public transportation is high.	5.327	1.341	−0.368	−0.522
CNT6 ^%^	To avoid crowded places in other provinces.	4.750	1.513	−0.653	−0.256
CNT7	Other people who are important to you (your family, friends) would not agree you go on long-distance trips.	4.704	1.502	−0.528	−0.260
CNT8 ^%^	Trust in the local government’s prevention policy encourages me to not travel to other provinces.	5.341	1.289	0.011	−0.575
CNT9 *^%^	I do not understand the prevention policy of other provinces.	5.346	1.307	0.199	−0.687
CNT10 ^#^	Less disposable income makes me travel in-state/province.	5.070	1.481	0.204	−0.739
Satisfaction with pandemic-restricted travel					
SA1	Overall, I am fully satisfied with the tourism experiences on this trip.	5.550	0.934	1.229	−0.520
SA2	Overall, I think it is value for money and time to take this visit.	5.510	1.024	−0.319	−0.255
SA3	Overall, the experiences I have had on this trip meet my expectation	5.517	0.997	0.449	−0.419
SA4	Overall, the level of satisfaction with this trip is high.	5.450	0.984	−0.384	−0.208
Intention of continuous pandemic-restricted travel					
IVD1	I would visit other pandemic-restricted travel destinations in the future.	5.115	1.159	−0.453	−0.133
IVD2	If given the opportunity, I would like to have other pandemic-restricted travel.	5.173	1.244	−0.319	−0.294
IVD3	I am loyal to this kind of pandemic-restricted travel.	5.481	1.070	0.632	−0.565

Note: * deleted items from EFA; # deleted items from CFA; % new items from the focus group interview.

**Table 3 ijerph-19-04149-t003:** The results of exploratory factor analysis (EFA).

Motivation KMO = 0.858, Variance% = 58.66			Constraint KMO = 0.848, Variance% = 57.127		
Factors	Exploration	Leisure	Factors	Perceived Risk	Policy Restriction
MO1	**0.800**	0.141	CNT1	0.137	**0.805**
MO2	**0.805**	0.204	CNT2	0.284	**0.695**
MO3	**0.775**	0.133	CNT3	0.187	**0.826**
MO4	0.406	**0.558**	CNT5	**0.663**	0.279
MO5	**0.753**	0.218	CNT6	**0.798**	0.174
MO6	0.387	**0.601**	CNT7	**0.685**	0.258
MO7	0.332	**0.605**	CNT8	**0.648**	0.269
MO9	−0.055	**0.791**	CNT10	**0.640**	0.027
MO10	0.129	**0.767**			

**Table 4 ijerph-19-04149-t004:** Reliability, construct validity and discriminant validity.

Dimensions	Cronbach’s Alpha	CR	AVE	Fornell–Larcker Criterion							Heterotrait–Monotrait Ratio (HTMT)			
				1	2	3	4	5	6	1	2	3	4	5
IVD	0.790	0.876	0.703	**0.838**										
EX	0.829	0.886	0.660	0.405	**0.812**					0.489				
LE	0.729	0.831	0.552	0.499	0.546	**0.743**				0.654	0.698			
PRN	0.760	0.847	0.581	0.337	0.352	0.412	**0.762**			0.426	0.434	0.548		
PRS	0.730	0.848	0.650	0.322	0.248	0.363	0.534	**0.806**		0.415	0.308	0.496	0.715	
SA	0.875	0.914	0.727	0.538	0.473	0.606	0.450	0.408	**0.853**	0.637	0.545	0.753	0.546	0.507

Note: IVD—Intention of continuous pandemic-restricted travel; EX—exploration; LE—Leisure; SA—satisfaction with pandemic-restricted travel; PRS—Policy restriction; PRN—Perceived risk for normal travel; CR—composite reliability; AVE—average variance extracted; Bold—square root of AVE (average variance extracted).

**Table 5 ijerph-19-04149-t005:** Structural equation modelling.

	Hypotheses	Path Coefficient (ß)	*p* Values	VIF	Status
H1a	Exploration → Satisfaction with pandemic-restricted travel	0.167	0.002	1.464	Support
H1b	Leisure → Satisfaction with pandemic-restricted travel	0.410	0.000	1.574	Support
H2a	Policy restriction → Satisfaction with pandemic-restricted travel	0.146	0.003	1.425	Support
H2b	Perceived risk for normal travel → Satisfaction with pandemic-restricted travel	0.138	0.018	1.477	Support
H3	Satisfaction with pandemic-restricted travel → Intention of continuous pandemic-restricted travel	0.310	0.000	1.803	Support
H4a	Exploration → Intention of continuous pandemic-restricted travel	0.109	0.032	1.514	Support
H4b	Leisure → Intention of continuous pandemic-restricted travel	0.206	0.000	1.877	Support
H5a	Policy restriction → Intention of continuous pandemic-restricted travel	0.062	0.272	1.464	No support
H5b	Perceived risk for normal travel → Intention of continuous pandemic-restricted travel	0.063	0.266	1.512	No support

**Table 6 ijerph-19-04149-t006:** Effect deconstruction of the structural model.

	Direct Effect	Indirect Effect	Total Effect	Confidence Intervals		VAF	Mediation
				2.5%	97.5%		
Exploration	0.109(0.032)	0.052(0.017)	0.161(0.003)	0.016	0.101	0.323	Partial mediation
Leisure	0.206(0.000)	0.127(0.000)	0.333(0.000)	0.073	0.185	0.381	Partial mediation
Perceived risk	0.063(0.266)	0.043(0.029)	0.106(0.081)	0.008	0.085	0.406	Full mediation
Policy restriction	0.062(0.272)	0.045(0.015)	0.107(0.069)	0.013	0.086	0.421	Full mediation

## Data Availability

Data from this study are available from the corresponding author on reasonable request.

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
