# Peer review of "An Exploratory Study of Pandemic-Restricted Travel—A New Form of Travel Pattern on the during- and Post-COVID-19 Era"

_ijerph, 2022, doi:10.3390/ijerph19074149_

Round 1
Reviewer 1 Report
The article presents an interesting study regarding constraints and motives to travel and the role of previous travel satisfaction in the intention to travel. Although the theme is interesting, major revisions are required.
The abstract should be improved. Please insert the main results in the abstract.
There are several corrections to be made
On line 82 Since COVID-19 has dramatically changed tourists’ physiological attitude [14], please change physiological with psychological.
Please insert a conceptual model depicting the hypothesized relationship between variables based on hypotheses.
Please insert relevant work
At line 84 Although researchers have suggested some possible tourism patterns [https://doi.org/10.3390/ijerph182111169], they have not considered the travel restrictions that influence tourist’s motivation and perception to travel [15].
Although COVID-19 vaccines are probably one of the best ways to help the tourism economy recover, it is not a “free pass” vaccination for travel depending on many factors [ https://doi.org/10.3390/ijerph19020918].
On line 99 Motivation is a phycological need change to psychological need
line 128 A Satisfaction change as Satisfaction
line 204 7-Likert scales change to 7-points Likert scales
Please correct items in Table 2 (for instance To travel to attracted by promotion)
In table 5 please change Not support to no support
The method section should be revised
Please introduce a 3.2 Measure subsection before data collection after the development of the instrument. Describe there the measures so it is easier to understand the measurement.
Please change Continuous pandemic-restricted travel as intention to travel. The items refer to subjective assessment of their intention, not of actual travel. Please do so throughout the entire manuscript.
please remove the from the phrase - Continuous the pandemic-restricted travel -
Please insert a section of exploratory or ancillary analyses to refer to mediation analysis. There are no prior hypotheses regarding mediation, so please specify- Please insert the conceptual framework based on hypotheses.
Please correct the text line 337-338 so as to have meaning On the other hand, two constraints to normal travel positively influence tourists’ 337 satisfaction with pandemic-restricted travel, but both do not directly influence intention 338 to take continuous pandemic-restricted travel.
on line During the pandemic, certain travel motivations, such as visiting friends, became unimportant. please cite https://doi.org/10.1080/10548408.2022.2045246 and reformulate as VFR seems important for COVID 19 tourism. MDPI will have soom an entire special issue on this.
line 390 please correct …..tourists’ phycological needs
5.2. Theoretical implication
Please remove the subsection and insert it after the discussion. The main part of the theoretical implications is the discussion of the results in relation to other studies. Summarize what is new and put it as theoretical implications.
5.4. Limitation and future studies
Please discuss the limitation of interpreting mediation based on simultaneous measures.
Please explain why Informed Consent Statement is Not applicable. Also, why Institutional Review Board Statement is Not applicable. This is contradictory with minimal research practice.
Reviewer 2 Report
The reviewed article was properly prepared in terms of literature review, hypothesis development, as well as the structure and sequence of sections. The article deals with very current and crucial issues related to the influence of the COVID-19 virus on the purchasing behavior of customers in the tourist market. It is a valuable attempt to identify the factors influencing the intentions of tourists to constantly visit a tourist destination due to COVID-19 limitations. The structural equation modelling (SEM) were used to verify the model between the selected variables.
Small suggestions only help with minor corrections and absolutely do not detract from the value and high quality rating of the reviewed article. Among the weaknesses of the presented text - in my opinion - should be indicated:
- It is also worth notice that the variable “Continuous the pandemic-restricted travel” included in the proposed model (table 2 p. 6 and figure 1 p. 9) concerns the buyers' declarations about future behavior, and not real behavior. In this situation better description of the variable is the "Intention of continuous the pandemic-restricted travel".
- There is lack of the following information:
- which method of sample selection has been used?
- explanation regarding the chosen sampling technique and the method of selecting the research sample;
- It is worth adding information about the data analysis method used, both to the abstract and to the keywords, and to clearly emphasize the advantages of SEM and the arguments for using this method.
- 1 p. 9 in the printout is illegible – its quality should be improved. Dla zwiÄ™kszenia przejrzystoÅ›ci i poprawy komfortu czytelnika można również wprowadzić symbole hipotez badawczych na fig. 1. p. 9
In my opinion, the article will be suitable for publication, however, after completing the above information and answers to the above questions. Thank you for the opportunity of reviewing this interesting article.
Reviewer 3 Report
Thank you very much for the opportunity to read the text.
I would like to highlight a few points that need to be improved. The indication that the author is for correspondence is placed after the surname and not before the first name.
Abstract should be supplemented with information on the period of the survey and the determination of the country in which it took place. What methods were used to develop the obtained results. Well, at the beginning, what was the purpose, in the abstract, the authors included only the results.
The font size in tables should be smaller.
The discussion is poorly written and needs redrafting. Discussion is the place where authors should confront their results with those of other scientists. The "Theoretical implication" sections could be considered for discussion - so I would advise you to leave these subsections in the discussion, and move the remaining three to another section - Conclusion. But there must be changes, it is not enough just to move redundant sections. All the results that confirm or contradict the hypotheses must be confronted in the discussion.
The number of literature items used is not large, so the authors can safely expand the discussion by referring to the literature on the subject, which is already expanded to include studies on the pandemic.
Good luck
Round 2
Reviewer 1 Report
The article includes adequate changes and specifications of the limitations.
Author Response
Thanks for your support.
Reviewer 3 Report
The Discussion chapter is still poorly structured.
There is no discussion as such.
And the Conclusion subsection is too long.
For example, it should be:
4. Discussion
4.1 Theoretical background
4.2 Future Research
4.3 Limitations
5. Conclusion
Author Response
Response:
Thanks for your comment. As we are not very familiar with the style and structure of the journal, we reviewed several updated articles in IJERPH (e.g. Ren et al., 2021; Ruan et al., 2021) and restructured these two sections as:
- Discussion
5.1 Theoretical implications
5.2 Practical implications
5.3 Limitations and future studies
- Conclusions
- Ren, Q., He, B., Chen, X., Han, J., & Han, F. (2021). The Mechanism and Mediating Effect of the “Perception–Emotion–Behaviour” Chain of Tourists at World Natural Heritage Sites—A Case Study from Bayanbulak, China. International Journal of Environmental Research and Public Health, 18(23), 12531.
- Ruan, W. J., Lee, J., & Song, H. (2021). Understanding Tourist Behavioural Intention When Faced with Smog Pollution: Focus on International Tourists to Beijing. International Journal of Environmental Research and Public Health, 18(14), 7262.
Hope this revised version can help improve the structure of the discussion chapter.